# Essential Per- and Polyfluoroalkyl Substances (PFAS) in Our Society of the Future

**DOI:** 10.3390/molecules30153220

**Published:** 2025-07-31

**Authors:** Rudy Dams, Bruno Ameduri

**Affiliations:** 1Rudy Dams Consulting, 2070 Zwijndrecht, Belgium; rudy-dams@outlook.com; 2Institute Charles Gerhardt, University of Montpellier, CNRS, ENSCM, 34296 Montpellier, France

**Keywords:** applications, closing the fluorine loop, environmental impact, essential uses, fluorochemicals, per- and polyfluoroalkyl substances, PFAS

## Abstract

Per- or polyfluoroalkyl substances (PFASs) are man-made compounds involved in compositions of many industrial processes and consumer products. The largest-volume man-made PFAS are made up of refrigerants and fluoropolymers. Major concerns for our society related to these substances are their contribution to global warming as greenhouse gasses and the potential for adverse effects on living organisms, particularly by long-chain perfluoroalkyl acid derivatives. Restrictions on manufacturing and applications will increase in the near future. The full remediation of historical and current contaminations of air, soil and water remains problematic, especially for ultra-short PFASs, such as trifluoroacetic acid. Future monitoring of PFAS levels and their impact on ecosystems remains important. PFASs have become integrated in the lifestyle and infrastructures of our modern worldwide society and are likely to be part of that society for years to come in essential applications by closing the fluorine loop.

## 1. Introduction

PFASs are a complex and intensely debated class of chemical compounds. The objective of this document is not to give a complete overview but rather to bring a broader perspective on this controversial topic in view of upcoming restrictions, particularly in Europe. Furthermore, the authors want to show the challenges to reaching a new global equilibrium between the worldwide applications portfolio of PFAS and their environmental, health, safety, regulatory and strategic implications. Last but not least, the authors want to emphasize that finding adequate fluorine-free alternatives and complete remediation (mineralization) techniques will require substantial R&D resources and implementation time and effort.

This document focusses on the scientific findings and debates on PFAS-applications and restrictions and proposes a solution based on essential uses, closing the fluorine loop and circularity.

## 2. The PFAS Family

Organofluorine compounds, of which PFASs are a special subclass, are synthetic organic substances that contain at least one C-F bond [1] (Figure 1).

PFASs are organic compounds consisting of carbon chains in which the hydrogen atoms are replaced by fluorine atoms (completely: perfluorinated, partial: polyfluorinated compounds). In line with the 2023 Organization for Economic Cooperation and Development (OECD) and European Chemicals Agency (ECHA) [2], PFASs represent fluorinated materials bearing at least one fully fluorinated methyl or methylene group, without any H, Cl, Br or I atom attached to it. With a few exceptions, any chemical with at least one perfluoromethyl group (-CF_3_) or perfluorinated methylene group (-CF_2_-) is regarded as a PFAS [2]. Other definitions have also been suggested [1,3].

A few examples of PFASs are CF_3_COOM (including TFA), C_7_F_15_COOM (including PFOA) and C_8_F_17_SO_3_M (including PFOS), with M being a hydrogen ion, a metal ion or an amine or ammonium salt, CF_4,_ HFC-125 (CHF_2_CF_3_), HFC-134a (CH_2_FCF_3_) or HFO-1234yf (CF_3_CF = CH_2_), and polymers, such as PTFE (based on -CF_2_CF_2_- units) or PVDF (with -CF_2_CH_2_- units) and (co)polymers containing TFE or VDF.

Some examples of non-PFASs (Figure 2) are HFC-41 (CH_3_F), HFC-32 (CH_2_F_2_), HCFC-22 (CHF_2_Cl), HCFC-142b (CClF_2_CH_3_), HFC-152a (CH_3_CHF_2_), CFC-11 (CCl_3_F), CF_3_I, NF_3_, SF_6_ or LiPF_6_.

Due to the very broad definition, the PFAS class contains over 14,000 individual compounds [1,2]. The PFAS members have a wide range of different and unique physical and chemical properties, from single organic molecules to high-molar-mass polymers (Figure 2). Therefore, great care must be taken not to make generalizations or oversimplifications based on missing data on individual products, such as considering the big class as equal under the same PFAS umbrella [4].

The vast majority of all PFASs (over 85%; estimated over 200 million ton) present in the Earth’s ecosphere consist of trifluoroacetic acid (TFA) and its salts [5,6,7,8]. The origin is controversial, but the degradation of man-made refrigerants, agrochemicals, drugs and fluoropolymer precursors is a significant contributor to the estimated yearly increase in TFA and its salts of a few thousand tons per year [5,6,7,8].

The remaining PFASs (estimated at less than 15%) consist of a big family of man-made organofluorine substances (mostly refrigerants and fluoropolymers; Figure 2) produced over the last 90 years and increasing fast by over a million tons/year [9]. Today’s man-made production and use of PFASs is therefore not sustainable in the (near) future.

Of the anthropogenic PFASs, the vast majority (about 65%) are non-functional and water-insoluble fluids and gasses, such as HCFC, HFC, HFE, HFO or PFC; many of these do not bioaccumulate in living organisms, but they are persistent by themselves or their degradation products.

About 15% are fluoropolymers [10] (they are only 38), such as PTFE, better known under its Teflon^®^ trade name from Du Pont (now Chemours, Wilmington, DE, USA), or PVDF, known as, for example, Kynar^®^ (from Arkema, Colombes, France) or Solef^®^ (from Syensqo, formerly Solvay-Solexis, Bollate, Italy). These polymers are extremely stable, non-toxic and, due to their high molar masses (and thus their size), non-bioaccumulative, and they do not cross the cellular membrane. Therefore, they can be considered as “*Polymers of Low Concern*” [11,12]. As an example, Henry et al. [11] reported a supplement of the ISO 10,993 data on PTFE in several forms (tubes, patches, fibers) evidencing criteria of PLC.

About 1–2% of all man-made PFASs, but increasing fast due to the breakdown of other PFASs, fall in the category of perfluoroalkyl acids, such as perfluoroalkyl carboxylic acids (PFCAs) (for example, PFOA) or perfluoroalkyl sulphonic acid (PFSA) derivatives, such as PFOSs [1]. Many of these compounds are of public concern, are water soluble (and thus mobile in aqueous media) and are bioaccumulative, especially those members which possess a long perfluoroalkyl chain of six or more carbon atoms [13].

Man-made PFASs are expensive to manufacture, resulting in a considerable environmental footprint, and therefore should only be used where other substances cannot provide the required performance or where they can deliver the same performance using much smaller quantities.

## 3. PFAS History

Although the properties of hydrofluoric acid were already known during the Middle Ages, its structure was not elucidated until 1771. The first organofluorine compound, benzoyl fluoride, was discovered in 1852 by the halogen exchange of a Cl- versus an F-atom. Fluorine gas (F_2_) was first isolated in 1886 by Moissan, who earned the Nobel Prize in 1906.

Due to the difficulty of handling HFA and F_2_, especially on a larger scale, it took until the beginning of the 20th century before organofluorine chemistry and its applications took off [13] (Figure 3).

(a) The first big wave of fluorinated compounds started in 1928 with the discovery of chlorofluorocarbons (CFC’s) as refrigerants by the General Motors Company [14,15]. They approached Du Pont to manufacture them on a large scale. Du Pont started the production of CFC-11, 12, 113 and 114 and HCFC-22 (better known under their brand name Freon^®^) in the 1930s. They also expanded the technology by using the CFCs as building blocks to synthesize fluorinated monomers such as TFE and hexafluoropropylene (HFP) (from HCFC-22). Today, HCFC-22, which is a non-PFAS according to the ECHA-definition, is still the biggest-volume man-made organofluorine compound (over 800,000 tons per year) [16,17,18,19].

Though the homopolymerization of chlorotrifluoroethylene (CTFE) was pioneered by IG Farbenindustrie AG [20], subsequently, the Du Pont discovery of PTFE in 1938 launched the fluoropolymer “revolution”. PTFE production started in 1949 [13].

(b) The second wave of organofluorine compounds started during World War II with the Manhattan Project [21]. ^235^UF_6_ enrichment, using HF and F_2_, needed fluorinated refrigerants, lubricants, valves, tubings, engineering devices and coatings stable regarding HF and F_2_, used in the production of the first atomic weapons.

As a result, Simons at Penn State University developed Electrochemical Fluorination (ECF) technology to make perfluorocarbon derivatives [22]. In 1947, 3M Company acquired the technology and started the development of commercial products, which came on the market starting in 1952 [23]. In 1950, Du Pont developed telomer chemistry, by a controlled oligomerization (named “telomerization” in presence of chain transfer agents) of TFE and end-capped by an iodine atom [24,25].

Both technologies allow for the preparation of functional perfluorinated building blocks [13], which became the most important starting materials for oil and water repellent coatings for textiles, paper, carpet, apparel or leather and surfactants for aqueous firefighting foams (AFFF). The main difference between ECF-derived and telomer products is that ECF delivers a mixture of branched and linear perfluorocarbon tails of mostly one chain length—for example, C_8_F_17_ segments—besides some shorter-chain by-products. Telomer products always contain a distribution of several linear chain lengths and thus a mixture of C4-C6-C8-C10-C12-C14 perfluoroalkyl moieties. Those characteristics make it easy for today’s analytical techniques to detect the fingerprints of each technology and distinguish the origin of the PFAS-contaminants.

After 1950, the US Air Force, Du Pont (and, later on, Unimatec), Montecatini/Ausimont and Daikin pioneered perfluoropolyether technologies using hexafluoropropylene oxide (HFPO)-oligomerization, the UV-photooxidation of TFE/HFP or the ring-opening of fluorinated oxetanes, respectively, leading to oligomeric materials containing single or combined repeating units of -CF_2_O-, -CF_2_CF_2_O-, -CF_2_CF_2_CF_2_O- and -CF(CF_3_)CF_2_O- segments, as well as mono- and difunctional or telechelic oligomers [25,26,27].

(c) Between 1950 and 1985, organofluorine and PFAS-technology and production flourished with an exponential growth in consumer and industrial products and applications. At the same time, the first negative aspects also appeared, namely, bioaccumulation and the environmental impact [13].

As a result, in 1987, the Montreal Protocol was implemented, banning the commercial use of CFCs due to their high Ozone Depletion Potential (ODP) as a result of the release of chlorine radicals in the stratosphere, attacking the ozone layer [28,29]. As a result of the ban, the third wave of new developments started with the discoveries of CFC-replacement products, such as HCFCs, HFCs, HFEs and HFOs. Even after the Kyoto Protocol in 1997 [30] to limit and reduce greenhouse gases emissions (including fluorinated ones) in accordance with agreed individual targets, this third wave is still ongoing today as the search for organofluorine compounds with zero ODP and minimum global warming potential (GWP) continues [31]. Actually, non-polymeric PFASs contribute to 94% emissions, while the contribution of fluoropolymers is estimated at 6%.

(d) Other aspects of concern arose in the 1990s [13] when the widespread presence of PFASs in the Earth’s ecosphere, including the blood serum of humans, became clear. As a result, 3M decided on a voluntary withdrawal from all markets using long-chain (C6-C8-C10 chains) perfluoroalkyl-containing products. A fourth wave of fluorinated materials started with the development of short-chain (C4-C6 perfluoroalkyl-containing) products by many commercial players, including China and India [32].

With an increased focus on the persistence and toxicity of PFASs in recent years, environmental and political pressure is building to introduce broader restrictions on PFAS manufacturing, applications and uses, especially in Europe [2].

## 4. Why Fluorine and Organofluorine Substances?

Fluorine is the 13th most abundant element in the crust of the Earth [16,33].

The Fluorine atom has special characteristics such as its size and its electronic surrounding. The fluorine atom has a very small Van der Waals radius, only 20% larger than that hydrogen, making it possible to sterically replace hydrogen in many hydrocarbon compounds. The fluorine atom also has a unique electronic environment, making it the most electronegative element in the Periodic Table, resulting in very strong C-F bonds. Furthermore, due to its three electron pairs, an “electronic shield” to its environment is built [13,26].

As a consequence, PFASs have exceptional physicochemical properties, such as outstanding chemical, thermal, electrical and biological stability for electronic, electrical, automotive, aviation and aerospace applications, as coatings in chemical and nuclear installations and for medical devices. Furthermore, they offer unique advantages in specific applications, such as

Low heat of vaporization for refrigerants,Low surface tension for surfactants,Low surface energy for oil, water, stain and dirt repellents for soft and hard surfaces,Low refractive index for optical applications,High solubility for oxygen for organ preservation and cancer therapy,High anti-stick and anti-friction properties for lubricants and release agents,Biocompatibility for medical devices and pharmaceuticals,UV-stability for many fluoropolymers.

In view of these outstanding properties, finding PFAS-free alternatives for the multiple uses continues to be quite challenging. Therefore, a Swedish team prepared a database listing 325 applications across 18 use categories [34]. Based on a screening of potential concerns, their performance compared to PFASs and their availability on the market, it was concluded that potentially suitable alternatives are available for 40 different applications. For 83 applications, no alternatives could be identified [34] and further studies were requested to find out real ones that are able to undergo the stringent requirements. Another study [35], based on Artificial Intelligence from more than 32,250 reports, patents, documents and publications, suggested 32 sub-families as potential replacements, encompassing metals, nanomaterials, composites and polymers. Among them, two could be identified as alternatives. However, the experimental testing and performance were not suitable. In addition, Jacobs et al. [36] also reported potential alternatives, but because of the combination of relevant features of fluoropolymers and many practical limitations, no alternatives were proposed.

## 5. Consumer Uses and Industrial Applications

The global market for organofluorine materials was valued in 2022 at about USD 25 billion [16,37], covering about 4–5 million tons of products per year. The biggest growth markets are air conditioning and refrigeration, transportation, aluminum production, pharma and agrochemicals, construction, and semiconductor and electronic applications [38].

In order to obtain a better idea on long-term sustainability, applications and uses will be divided into four groups: healthcare, food security, energy systems and materials.

### 5.1. Healthcare Applications

Since the registration of the first fluorinated pharmaceutical in 1954, over 340 fluorinated medical drugs, including several blockbusters, owe their activity to the presence of fluorine in their structure, favoring a better interaction with proteins and fats [39,40].

In the total medicines market, fluorinated drugs account for about 20% and ca. 5% are PFASs. In the last decade, over 15% of all new medicines have been PFASs [39,40]. The potential is expected to increase in the future due to advancements in synthetic strategies to incorporate -CF_3_, -OCF_3_ or -SCF_3_ groups [33,34] as well as SF_5_ (not considered as a PFAS) in organic compounds [39].

Important healthcare applications for PFASs include medical devices (such as implants, stents, tubings, catheters, artificial veins, cardiovascular prostheses and the like [41,42,43]), propellants for metered dose inhalers and anesthetics [44], perfluorocarbon emulsions [45] as well as organ preservation and advanced cancer treatments [40] and contrast agents [46].

### 5.2. Food Security

Since 1963, over 420 fluorinated agrochemicals have been commercialized as herbicides, fungicides, insecticides or acaricides. About 50% of the new agrochemicals are fluorinated, 70% of which represent PFAS. [47] The biological efficacy for crop protection and public health security against parasitically transmitted (by e.g., mosquitos) infectious diseases are the drivers for the success of fluorine in agrochemicals [39]. In the near future, restrictions prohibiting the excessive use of PFAS-pesticides are expected in Europe.

The European (European food safety authority, EFSA) and U.S. health regulation agencies have all confirmed that PTFE is not hazardous for health and is considered as an inert compound [48].

### 5.3. Energy

The continuous supply of sufficient and cheap energy for a growing world population is one of the most important challenges of the 21st century. Fluorine chemistry and PFASs now already play an important role by providing fluoropolymers ionomers (Figure 4) [49,50], gasses and fluids and electrolytes [49,50,51] in nuclear, oil and gas and membrane applications, lithium battery components (e.g., separators and binders for electrodes [51]), electrical and renewable energy applications (e.g., backsheets of photovoltaic panels, lubricants for turbines and aids to demold blades of wind mills) and piezo-or ferroelectric polymers [52]. It is expected that the dependency on PFASs in energy generation will continue in the next decades, although at the same time, the search for PFAS-alternatives is increasing fast [53]).

### 5.4. Materials

#### 5.4.1. Fluoropolymers and Oligomers

Since the discovery of PTFE, a broad range of fluoropolymers have been developed and supplied, finding applications in a wide range of technical fields where security, reliability and specific requirements are needed. Indeed, fluoropolymers are regarded as specialty (or high-performance) polymers.

The uniqueness of fluoropolymers is due to the excellent chemical, aging and thermal stability of highly fluorinated PFAS polymers in high-demanding operations such as automotive and transportation, aviation (e.g., hundreds of kilometers of cable are coated by PTFE since its Limit Oxygen Index, LOI, is quite high, 95, to favor a low generation of smoke and is not flammable), aerospace, telecommunications, oil and gas, the chemical and nuclear industry, healthcare, construction materials and specialty coatings [10,26,38,54].

The total market value is about USD 8–10 billion (2023) spread out over ca. 400,000 ton/year [55] and is expected to grow at a compound annual growth rate (CAGR) of 6% in the next decade. The family of fluoropolymers [56] is built around homo- and copolymers of TFE, HFP, VDF, CTFE, perfluorovinyl ethers and various cure-site monomers. Major members are PTFE with 45% [38], PVDF with 15% and elastomers with about a 10% share [57].

Perfluoropolyether (PFPE) oligomers contain -CF_2_O-, -C_2_F_4_O- or C_3_F_7_O-ether segments and are mostly used as high-performance fluids (hydraulic fluids) or lubricants (in fan bearings in tunnels, cars and wood panel industries) that are able to withstand extreme operating temperatures and pressures (NSF ISO 21469 certified) such as in air- and spacecraft engines and specialty machinery [26,27].

Interestingly, they are also food-approved. No alternative has been found in essential applications.

#### 5.4.2. Perfluoroalkyl (Meth)Acrylates

In contrast with fluoropolymers, where almost all fluorine atoms are located in the polymer backbone, designed low surface energy polymers have fluorine atoms present in the side chains. These side-chain fluorinated polymers (SCFPs) provide oil, stain, dirt and water repellency to a broad range of soft (e.g., textiles, apparel, paper, leather and carpet) and hard (e.g., ceramics, glass, concrete, stone and wood) surfaces [24,58]. Figure 5 represents the overall chemical structure of such copolymers, where each unit has its own function. Although the market for oil and water repellents is a specialty market, worldwide volumes were once ca. 20,000 tons of products per year, now declining and being replaced by fluorine-free repellents.

#### 5.4.3. Perfluoroalkyl-Derived Surfactants and Emulsifiers

Perfluoroalkyl surface active agents display very low surface tension in aqueous and organic media. They are extremely stable regarding acids, bases and oxidizing media and high temperatures, being surface active at very low concentrations. Therefore, these unique materials are involved in more than 200 applications, e.g., as emulsifiers for fluoropolymer manufacturing (though most recently, manufacturers use hydrocarbon surfactants [59]), as essential components in fire-fighting agents (AFFF and ATC) [60] (though fluorine-free alternatives exist [61], which can extinguish fires but do not meet the strict technical requirements for critical applications, such as military applications) and as wetting, leveling and spreading agents in aqueous and solvent-based paints and coatings, metal plating, electronics and semiconductor manufacturing [62].

#### 5.4.4. Fluorinated Gasses and Fluids

Organofluorine gasses and fluids amount to about 2 million tons per year, of which ca. 0.9 million tons are PFAS, making them by far the biggest PFAS market.

Major applications [17,18,19] are in refrigeration (therefore, the product codes start with an R), air conditioning and cooling (R-134a, 143a, 125, 22 and their blends; R-1234yf), blowing agents (R-141b, 134a, 365 and HFC-245fa), propellants and aerosols (R-227, 134a), fire extinguishing agents (R-227, Novec^®^ 1230), semiconductor fabrication (C_2_F_6_, C_3_F_8_), magnesium manufacturing (SF_6_), specialty solvents (HFC-4310mee and HFE-7100) and chemical feedstocks (R-22, 142b). The largest market, by far, is Asia Pacific.

In the last decade, the market has been moving away from HCFC and HFCs to no-ODP and lower-GWP materials (HFO, such as 2,3,3,3-tetrafluoropropene, R-1234yf) and fluorine-free materials (mostly hydrocarbons) [18,19,63,64,65].

## 6. Impact of PFASs on the Planet and Society

### 6.1. Unique Properties Leading to Environmental, Health and Safety (EHS) Issues

In the last three decades, it has become more evident that the unique properties of PFASs also lead to significant environmental, health and safety concerns [13] such as

persistence (long environmental and atmospheric lifetime),bioaccumulation potential in plants, animals and humans,potential for adverse effects on living organisms,mobility in aqueous systems and the atmosphere over a long range,absorption on soil,high GWP.

### 6.2. Environmental Aspects—Atmospheric Impact

Since 1800, the average temperature on Earth has increased by about 1.2 °C, mostly due to human activity. If no action is taken, the temperature on our planet will rise by over 2.5 °C by 2050, resulting in significant climate changes.

Some organofluorine compounds, including PFASs, have long (years to decades for HFC) to very long (millennia for PFCs, SF_6_ and NF_3_) atmospheric lifetimes. In combination with their strong IR absorbance, they are very potent greenhouse gasses [63], which the co-authors believe to be the most important PFAS-related concern.

The total atmospheric concentration of fluorinated compounds has grown rapidly since the 1950s but has started to level off since 2019 (the Kigali amendment to the Montreal Protocol). Nevertheless, currently, about 800 ppt of CFC-12, 500 ppt of HCFC-22 and 100 ppt of CF_4_ are present in the atmosphere [64], and emissions stay significant. Indeed, several million tons per year are released by the electronic industry (CF_4_, NF_3_ and PFCs), the electrical industry (SF_6_) and aluminum manufacturing (CF_4_) as some important sources. Fluorinated materials make up about 3% of all greenhouse gasses in the atmosphere [63].

### 6.3. Environmental Aspects—Impact on Water and Soil

Many PFASs, as well as their breakdown products, do not degrade easily [66,67,68] and therefore accumulate over time over the globe and its ecosphere, resulting in the widespread contamination of soil and water, primarily by PFCA derivatives (mostly C2, C3 and C4 PFCA salts, but going up to C16 analogs) and some C4-C6-C8 PFSAs (such as PFOS).

Some PFASs break down quite fast [69,70]—for example, HFO-1234yf in a couple of days or N-ethyl perfluorooctyl sulphonamido ethanol (EtFOSE) in weeks under aerobic conditions, unfortunately leading to other persistent PFASs. On the other hand, some PFASs degrade slowly (e.g., phosphate ester water and oil repellents), requiring years, degrade very slowly (oil and water repellent polyacrylates), requiring decades to centuries, or degrade to almost nothing at all (fluoropolymers, PFOA and PFOS at room temperature) [38,68,71,72].

### 6.4. Remediation Technologies

Remediations for PFAS contaminations should consist of three essential steps: capturing/removal, mineralization into fluorides and recycling of the fluoride salts. Remediation technologies are needed to bring not only solutions for the historical contaminations but also for the current worldwide emissions to air, soil and waters.

Typically, PFAS removal from groundwater relies on absorption techniques (Figure 6) using conventional sorption materials such as granular activated carbon (GAC) and ion exchange resins, or on filtration or separation methodology such as ultrafiltration or reverse osmosis (RO) [73,74,75]. For long-chain PFAS (C6 and higher), these capturing techniques work well, but for ultra-short (C2–C3) homologs, they are inefficient and often ineffective [74], except for RO. Furthermore, inorganic salts, particles and organic matter interfere (strongly) with the absorption and separation processes.

Remediations for contaminated soil quickly reach their limits, since excavations and subsequent decontaminations require huge effort and are quite challenging due to the lack of uniform standards for recycling and disposal, the shortage of clean replacement soil and, again, the unavailability of efficient capturing techniques for ultra-short PFAS. Overall, soil washing and soil stabilization/immobilization have progressed to be the most established and most implemented technologies for PFASs, with an increasing number of full-scale tests. While these technologies can be highly effective and pragmatic, residual PFAS concentrations may not achieve the most stringent regulatory thresholds [76]. Furthermore, high costs estimated at USD 20 to 7000 trillion per year for the removal of perfluoroalkyl acids (a subclass of PFASs) [77], and the uncertainty of the true origin of the widespread contamination, make remediation in practice very complicated.

Up to now, decomposition for most PFASs have been achieved economically by heating up to 1400 °C for a minimum of few seconds in special ovens [78], such as cement kilns. Incineration is the best overall technique that is available, although in practice, challenges remain for complete removal [79], as recently reported by Gehrmann et al. [80] at high temperatures.

New remediation techniques (Figure 6) are under development [81,82,83,84,85], even reaching the pilot plant stage (e.g., photocatalysis, supercritical water oxidations, non-thermal plasma, electrochemical, sonochemical, high-temperature alkaline treatments, bioabsorption by plants followed by incineration of the leaves and ball milling). In a very interesting state-of-the-art document, Pilli et al. [86] comprehensively reported the tools and techniques for the remediation of PFASs in the environment. They also reported new economical and sustainable treatment methodologies that are valuable in removing PFASs from the environment across the world (Figure 7).

More recently, Alnaimat et al. [87] underlined electrooxidation (EO) as a key solution for pollutant degradation. This study significantly contributes to the advancement of PFAS degradation technologies, proposing a reliable tool for environmental remediation features.

However, no cost-effective solution for complete remediation (meaning the capturing, full mineralization and recycling of the formed fluoride) with a 100% closed fluorine loop in a short residence time (minutes) is available yet, especially for ultra-short PFASs, at industrial flow rates and in the presence of other contaminants, such as inorganic salts, heavy metals, soaps or other domestic waste streams. Certainly in situ solutions are missing, as well as continuous processes for PFAS removal from surface waters, groundwater, soil, sediments and biota. Combined treatments with complimentary effects may bring better solutions in the near future [81,82,83,84,85].

Related to the destruction of polymeric PFASs, though a review reports both the thermal depolymerization and mineralization of FPs [88], Gouverneur’s group [89] has recently demonstrated an efficient method, under a simple mechano-process in the presence of K_3_PO_4_, for totally mineralizing various PTFE and PVDF samples (including films, tubes and other items), thus enabling the fluorine recovery as KF and K_2_PO_3_F.

### 6.5. Reduction in Impact by Industry

Since the first significant signs of negative effects of PFASs in the early 1980s [13], industry has taken actions to reduce its environmental footprint.

Some important steps were

Worldwide emissions to air and water by fluoropolymer production [26,90,91] were significant in the past decades. In the 1990s, Du Pont and Hoechst started recycling APFO, the ammonium salt of PFOA, used as an emulsifier in fluoropolymer production, and replaced it initially with other non-bioaccumulative fluorosurfactants [26] and later by fluorine-free polymerization aids [59]; the recycling and re-use of fluoropolymers took off [26,91]. Air emissions have been drastically reduced by many producers, but further worldwide reductions are mandatory [88,92].2000: 3M initiated the voluntary phase-out of C6, C7, C8 and C10-based PFAS products and their manufacturing and the company announced it will stop PFAS manufacturing by the end of 2025 [93].During the last two decades, more than 100 fluorine-free alternatives for oil and water repellents, AFFFs, paint and coating additives, cosmetic additives and food packaging have been developed and marketed [94], although not always successfully, since most replacements are not drop-in solutions abd are still in the R&D stage. Their supply chain is not secured yet or cannot match the performances and cost of their PFAS counterparts [34,35].

### 6.6. Toxicological Aspects

The widespread use of PFASs and their release into surface waters and the air and leaching into soil and groundwater has become an important concern for our industrial society. Dietary intake by contaminated drinking water and food supply has become the predominant source of human exposure to PFASs [95].

(a) Bioaccumulation refers to the accumulation of a substance in an organism over time. Many PFASs exhibit high stability and lipophilicity (or proteinophilicity) and, depending on their chemical structure, the length of the perfluoroalkyl chain and its branching, prove to have remarkable bioaccumulation in all living species, including the human body, and limited excretion by the kidneys [96,97]. PFASs can bind to human serum albumin and other transporters in the blood [98]. The lungs demonstrated higher levels of PFASs accumulated compared to the liver, kidneys and brain tissues. Interestingly, Gui et al. [99] reported an association between PFAS exposure and the risk of diabetes. PFOS was found to be more prevalent in the liver, while PFCAs were predominantly detected in bone [96]. Additionally, there is a sex difference, with PFOS being excreted faster in females compared to males, although the difference in the elimination of PFOA is marginal [98]. The time for 50% reduction by excretion (mostly urinal) in humans for PFOS is 4–5 years, for PFOA, it is about 2 years, for PFBS, it is 26 days, for PFBA, it is 3 days and for TFA, it is about 16–26 h [100].

(b) Animal data are still commonly used to study toxicological effects and to set guidelines for drinking water and dietary intake. However, significant problems in extrapolating health effects from laboratory animals to humans in the context of PFAS exposure remain [100].

In rodents, PFASs appear to exert toxicological effects by binding to and activating the peroxisome proliferator receptor alpha, although the liver toxicity in rodents cannot be extrapolated to assess human health implications. Furthermore, the effects observed occurred only at PFAS concentrations several orders of magnitude higher than the mean PFAS concentration in the blood of Western populations [96]. Indeed, as far as acute toxicity is concerned, the no effect level (NoEL) for PFOS and PFOA in rats is about 50–100 ppm. The first negative effects observed in rats for PFOS are changes in the cholesterol levels, decreased immunity, an increase in liver weight and size and adverse reproductive effects [95].

Full toxicological data are available for just a few PFASs, such as TFA, PFOS or PFOA [101,102].

(c) The toxicological behavior of a compound in humans depends not only on its intrinsic potency, concentration and exposure time but also on individual factors such as age, sex, genetics and health status.

Epidemiological studies [102] have revealed statistical associations between exposure to specific PFASs (PFOS, PFOA) and certain triggers or biomarkers that (may) result in health effects in humans, such as altered immune systems, decreased thyroid function, decreased cholesterol levels, decreased or postponed fertility or kidney/testicular/liver diseases. The harmful effects may occur even at very low concentrations. The United States Environmental Protection Agency (EPA) has classified PFOA as a “*likely carcinogen*”, while PFOS is classified as “*possibly carcinogenic*” for humans [103]. More recently, Zahm et al. [104] reported that PFOA induces epigenetic issues and causes uterine adenocarcinoma and pancreatic alterations.

Chronic toxicity and the underlying mechanisms for PFAS are still not very well understood. Medical follow-up for industrial workers over 40 years has shown no widespread negative health effects, such as the occurrence of cancers, at current levels (e.g., risk for hormone receptor-positive breast cancer [105]). In highly exposed communities, reports have been made about reproductive and immunological disruptive properties [106,107,108]. Biggeri and Faccido [108] revelaed an association for kidney cancer and testicular cancer. A large pollution of surface, ground and drinking water contamination with PFASs was found in three provinces of Northern Italy, as well as contaminated sites in Germany [109].

Moreover, related to serum (mostly PFOS, PFOA, PFBA and PFNA) [110,111] the general population in Western society (e.g., Italy, Germany, the Netherlands [91], Belgium [111] and France [90]) has about 5–100 ppb PFASs in their blood serum, whereas industrial workers with high exposure carry heavier loads of 50–1000 ppb on average.

More recently, Hamscher [112] reported that, in 2008, the tolerable weekly intakes of PFOS and PFOA were 150 ng/kg body wt and 1500 ng/kg body wt, respectively, while in 2018, these values declined to 13 and 6 ng/kg body wt, respectively [113].

(d) Comparing the present and future concentrations of TFA [114], the dominant PFAS in the ecosphere, the threat for irreversible accumulation is rising [115], but risk for the environment and living species is anticipated to be under control for now [114,116]. However, a recent study raises concerns since malfunctions in rat offspring were found [117]. Therefore, future monitoring of PFAS and TFA concentrations and their impact on the total ecosphere remains very important, especially for drinking water and food.

As far as fluoropolymers are concerned, Naftalovich reported [118] that rats fed a diet of 25% PTFE for 90 days had no signs of toxicity.

### 6.7. The Analytical Dilemma

The PFAS family contains large amounts of over 14,000 compounds with very different properties. PFAS analysis in environmental samples is currently mainly caried out by liquid chromatography–tandem mass spectroscopy (LC-MS/MS) and related techniques as well as gas chromatography–mass spectroscopy (GC-MS) to conduct target analysis. This technique allows for the measurement, at very low levels (parts per trillion or even lower), of specific target compounds but suffers from high experimental errors (especially for ultra-short PFASs), poor recovery factors and long cycle times. Furthermore, only about 50 PFASs can be measured. Nevertheless, this technique is often used for the legal monitoring of PFASs present in environmental samples. Non-target high-resolution mass spectroscopy (HRMS), capable of measuring about 500 individual compounds for which no reference materials exist for now, is a logical extension.

To obtain a better view on the “total” amount of PFASs, sum parameter methods like total oxidizable precursor assay (TOPA), which allows for the determination of 60–75% of all PFASs, absorbable organic fluorine (AOF), extractable organic fluorine (EOF) and combustion ion chromatography (CIC), allowing 80–95% of all PFASs to be measured, are in advanced development. These methods do not give information on individual components (except for TOPA) and are not as sensitive as LC-MS/MS, although the equipment cost and cycle times are important advantages.

An important gap in analytical methods for individual compounds and sum parameters, as well as sensitivity and cycle time (in situ measurements), remains. In view of future restrictions or even bans on PFASs and their legal consequences, accurate standardization and improving PFASs’ individual and sum parameters are required in the very near future [119].

## 7. Closing the Fluorine Loop and Circularity

Some PFASs are known to be harmful to both the environment and human health, but others, such as fluoropolymers, are still under investigation [11,12,13,38,48,92,113,118,120].

On 7 February 2023, the ECHA published a proposal to restrict the entire class of PFASs [117]. The proposed restrictions are unprecedented and would affect all PFASs. The restrictions would only apply in Europe, but, in our opinion, since PFAS contamination does not stop at borders and the fate of imports of PFAS-containing products into Europe is still unknown, the restrictions may not stop PFAS contaminations in Europe and become a disruptive economic and strategic burden.

It is likely that the restrictions will not be fully implemented to all PFAS and their applications, for economic, strategic and social sustainability reasons. Derogations for certain sub-classes (for example, certain fluoropolymers) or specialty applications (e.g., medical uses, electronics or transportation) [38,96,121] may be allowed. Therefore, in our opinion, finding a new worldwide balance between the unique properties of PFASs and the environmental footprint on our global ecosphere and society is an absolute must (Figure 8).

This new worldwide balance comprises at least three steps:

(i)Focus on essential PFAS uses [37,122] and applications and phase out all others as soon as possible. By “essential uses”, it is meant those applications which are critical to our society and for which no viable and equally performing substitute exists.Examples of non-essential uses therefore include, for example, the packaging of popcorn and fast food, cosmetics and skin care, ski waxes, oil and water repellents for carpet, textiles, paper, home fabrics and the like, frying and cooking pans, window cleaners and paint leveling agents. On the other hand, examples of essential uses [122] where no adequate alternatives are available [35,36] for the time being, are some anesthetics, hydraulic fluids, optical fibers, lithium or sodium battery electrolytes and binders of cathodes, coatings and specific items for the nuclear and chemical industries, bridge and building bearings, semiconductor and chip manufacturing, high-temperature sealing applications, medical devices (implants, catheters, tubings, wound dressings and propellants for inhalers), insulations for cables, high-voltage switch gears, fire extinguishing for aviation, ships and military, fire regulated air conditioning, specialty refrigeration, professional clothing for army, police and firefighters and metal plating surfactants.(ii)Implement circularity [26,88,89,90,91,92,93,94,95,96,97,98,99] and closed applications for these essential uses, without burdening the environment (Figure 9). Environmental awareness requires considering the complete life cycle of products and consequently recycling PFASs, as exemplified for fluoropolymers by Dams and Hintzer [26] and in Figure 9.A key action will be to enforce new business models such as the licensing or leasing of PFASs (and chemicals in general), so producers stay accountable for their products.(iii)Strive for global zero-emissions in local manufacturing and applications, resulting in 100% mass balances.Key actions will be to implement remediation strategies for historical contaminations in water and soil as well as the complete mineralization of all emissions to soil, water and air and recycling the released fluorides [88,89]. Furthermore, creating a worldwide equilibrium between observed acute and long-term environmental and toxicological effects, analytical detection limits, global standards for contaminations, food and drinking water guidelines and, last but not least, overall norms and standards for industrial activities is crucial.

## 8. Conclusions

Organofluorine compounds and more specific PFASs, known as “*forever chemicals*” due to their persistence over decades, are part of the lifestyle and infrastructures of our modern society.

As a result of concerns over the worldwide spreading in air, soil and water and over the long-term health effects on living species in the ecosphere, restrictions on their use are coming (e.g., for ski waxes, food packaging or cosmetics) or are under review (e.g., for textiles, upholstery, leather, apparels, carpets, food contact materials, petroleum and mining applications). Many EHS&R-activities are still going on, even revisiting definitions of PFASs [3]. So far, research and regulatory agencies have primarily focused on the environmental occurrence and health effects of non-polymer PFASs, particularly perfluoroalkyl acids and their precursors. Quite recently, the EPA announced that the thresholds for PFOA and PFOS in water will be maintained, while those for the other PFASs will be deleted. In Europe, another consultation will be published in the coming months by the risk assessment committee (RAC) and the socio-economic assessment committee (SEAC) [123] after examining PFAS uses, e.g., as medical devices, as lubricants, in transportation, energy, electronics and semiconductor applications and in consumer mixtures. Some derogations are suggested, allowing for the continued manufacturing and use of PFASs [124] useful for batteries, membranes for fuel cells and electrolyzers made from fluoropolymers.

Therefore, it is believed that certain PFASs, and especially fluoropolymers, will remain part of our society for critical applications for years to come [55]. It would be hard to imagine returning to a society without safe, reliable and affordable air conditioning, medicines and medical devices, electronics, renewable energy and effective coatings, seals and tubings.

However, concerns over the sustainability of PFASs and applications must be taken seriously by focusing on (i) essential uses and phasing out all the rest in the near future, (ii) the complete remediation of contaminations and (iii) closing the fluorine loop in manufacturing and applications.

More research and development efforts are absolutely needed on knowledge gaps such as the environmental and toxicological impacts of individual PFASs and their mixtures, full remediation (mineralization) technologies and accurate analytical methodology focusing on individual PFASs but even more importantly on total organofluorine levels (including PFAS) and fluorine-free alternatives [34,35], where possible.

## Figures and Tables

**Figure 1 molecules-30-03220-f001:**
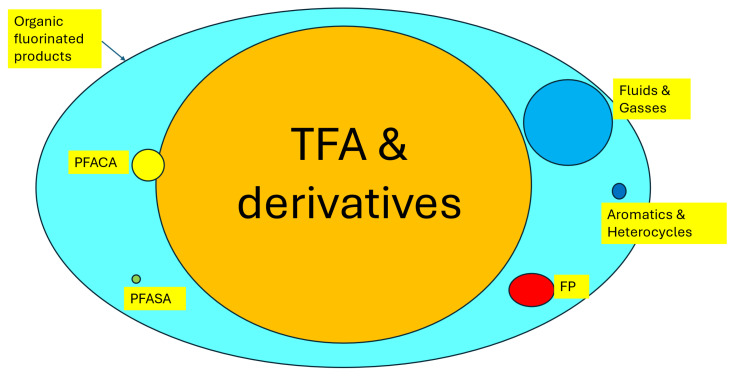
Overview of organofluorine compounds.

**Figure 2 molecules-30-03220-f002:**
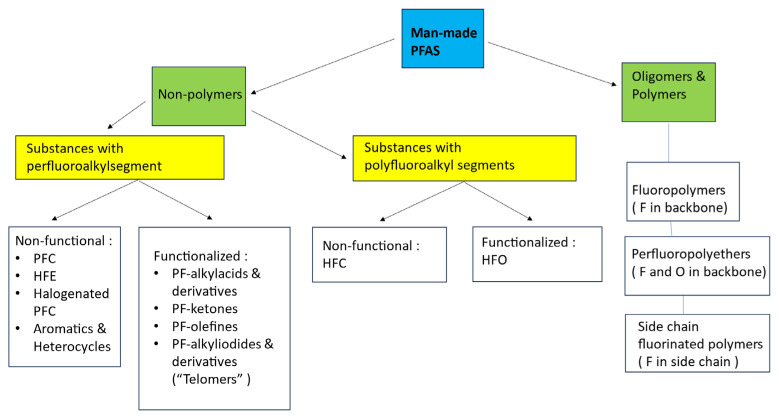
Overview and categorization of man-made per- or polyfluoroalkyl substances (PFAS), inspired by [1].

**Figure 3 molecules-30-03220-f003:**
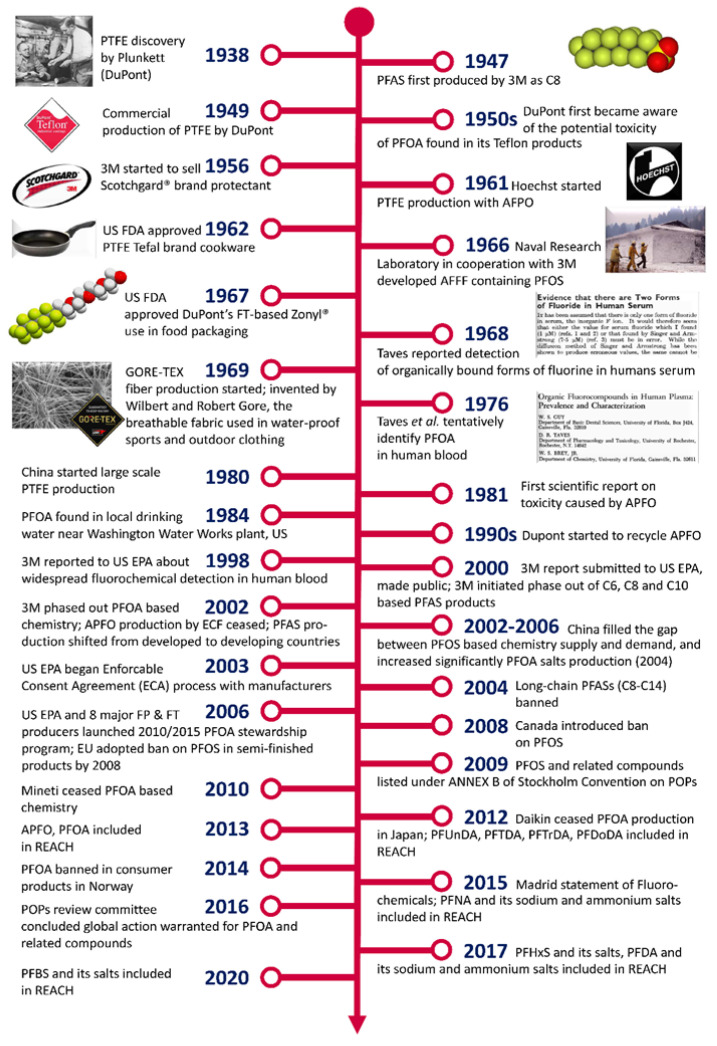
Brief history of the fluorinated compounds from the discovery of polytetrafluoroethylene (PTFE) to REACH regulations (APFO, ECF, EPA, FT, PFBS, PFCA, PFNA, PFOA, PFOS, POP and REACH stand for ammonium perfluorooctanoate, electrochemical fluorination, Environment Protection Agency, fluorotelomer, perfluorobutanesulfonate, perfluorocarboxylic acid, perfluorononanoic acid, perfluorooctanoic acid, perfluorooctanesulfonic acid, ersistent organic pollutants and Registration, Evaluation, Authorisation and Restriction of Chemicals, respectively (reproduced with permission from RSC, from [13])).

**Figure 4 molecules-30-03220-f004:**
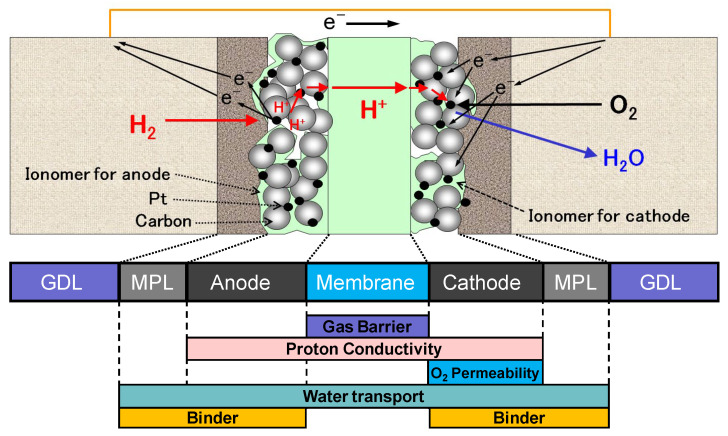
Sketch of a membrane–electrode assembly and ionomer functionality [49].

**Figure 5 molecules-30-03220-f005:**
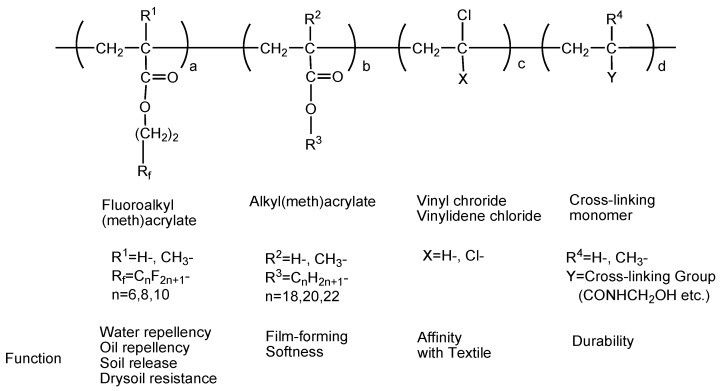
Copolymer based on fluoroalkyl (meth)acrylate and other complementary comonomers for specific features in the treatment of fabrics and textiles (e.g., water and oil repellents) [24].

**Figure 6 molecules-30-03220-f006:**
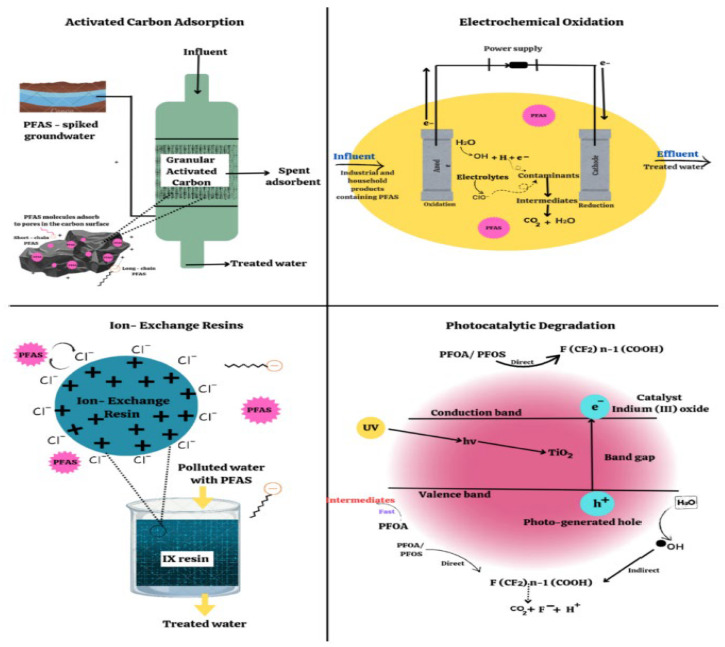
Non-exhaustive emerging treatment technologies used to remove PFAS contamination. Activated carbon adsorption (widely used technique for eliminating PFASs from drinking water); ion exchange (a more rapid reaction process requiring shorter contact periods); photocatalytic degradation (a sophisticated method of oxidation which harnesses light energy to break down contaminants through direct or indirect oxidation); electrochemical oxidation, a process where hydroxyl radicals, generated by anode material and electrolytes, oxidize PFASs, inducing its decomposition (reproduced with permission from Elsevier [75]).

**Figure 7 molecules-30-03220-f007:**
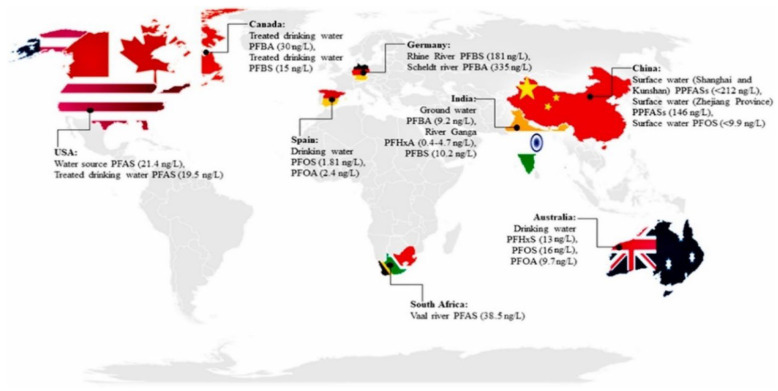
PFAS concentrations across the world (reproduced with permission from Elsevier [86]).

**Figure 8 molecules-30-03220-f008:**
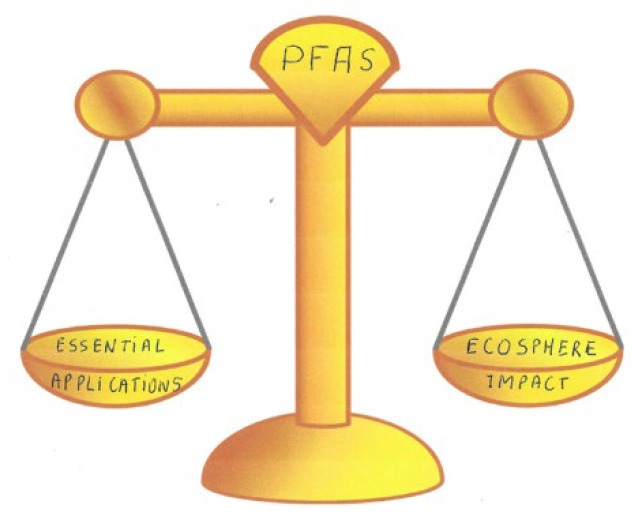
A worldwide PFAS-balance.

**Figure 9 molecules-30-03220-f009:**
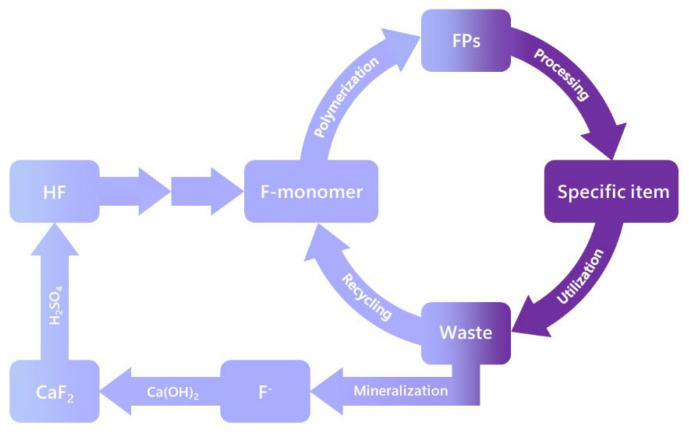
Life cycle and circularity of fluoropolymers.

## Data Availability

No new data were created since review and perspective article.

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
