# Peer review of "Essential Per- and Polyfluoroalkyl Substances (PFAS) in Our Society of the Future"

_molecules, 2025, doi:10.3390/molecules30153220_

Round 1
Reviewer 1 Report
Comments and Suggestions for Authors
attached

Author Response
Professor Bruno AMEDURI is well known scientist working in the area
of fluoropolymers.
This paper is a logic continuation of his reviews published before.
It is well-structured, well-written work which can be published without
alteration. Possibly, it is worthy to add recent information on fluoro-containing
polymers used in proton- and anion exchange membranes (C. Bae, P.
Jannasch and others.).
Answer: we appreciate these comments and, as the reviewer read, the initial manuscript cited 123 references (including also on membranes in reviews cited in ref. [49-50].). Hence, it is not worth adding further references to proof our point on unique uses and applications. Acually, we already published one review on protonic F-polymers for fuel cell membranes (Progr Polym Sci ; 2005, 30, 644-687) and another one on alkaline fuel cell membranes (Progr Polym Sci., 2011, 36, 1521– 1557). Section 5.3 in page 12 is devoted to « Energy » and displays Figure 4 as fuel cell membrane sketch, highlighted by ref. 49-50.
Reviewer 2 Report
Comments and Suggestions for Authors
Dams etal reported concerns over the sustainability of PFAS products and applications should be 35 resolved by focusing on essential uses, closing the fluorine loop and remediation 36 technology for historical contaminations. The authors cites a lot of literatures to show their considerations. It is well written, but its depth did not reach a high level. So it cannot accepted at the current state. Some comments should be provided:
- Too many common descriptions on the questions of PFAS. More science and technology topics should be provided. Some typical samples should be shown. For example, what is the most used one? It is the molecular structures? Why is it unique?
- Most PFAS should give some molecular structures.
- In order to reduce the use of PFAS, what is its substitutes? These substitutes is better in environment, or their disadvantages over PFAS. Some suggestions should be given for the development of PFAS’s substitutes
- The formation of the literatures should be paid attention, such as reference 84.
- Figure 9 is not a key for the whole paper. So it should be deleted.
- Is closing the fluorine loop is a good solution for PFAS? The topic need more deep discussions. In my opinion, it is only a solution, not only ones.
Author Response
- Too many common descriptions on the questions of PFAS. More science and technology topics should be provided. Some typical samples should be shown. For example, what is the most used one? It is the molecular structures? Why is it unique?
Answer 1 : many PFAS examples are included within the MS including structures; we explained why fluorine is unique (section 3). Related to the question on « which is the most used ? », indeed this is a case by case : for surfactants or fire fighting foams, amphiphilic short chain-organic derivatives are used while for specific materials (in medicine, aerospace, vehicles, optics or coatings, polymeric materials are requested.
2. Most PFAS should give some molecular structures.
Answer 2 : structures are given in section 2, but our point is not about chemical formulas (given in many cited references), but about finding a balance between PFAS properties and their environmental impact. Figure 5 supplies the chemical structure of a copolymer based on fluoroalkyl (meth)acrylate that is quasi not cited in other reviews related to PFAS. Therefore, we used limited chemical structures which can be easily found in the literature, but focused on our essential applications / closing the fluorine loop message. That is our point and we do not need chemistry to explain that point.
Q3: In order to reduce the use of PFAS, what is its substitutes? These substitutes is better in environment, or their disadvantages over PFAS. Some suggestions should be given for the development of PFAS’s substitutes
Answer 3 : throughout the whole MS, we have given examples of PFAS and possible alternatives (pages 10-14, 19, 22 and in conclusion) for textile treatment, refrigerants, AFFF mostly through literature references. Indeed, there is only little added value by repeating what is already published.
As for fluoropolymers, so far, no alternatives have been found (also specified in bottom of page 10) as well-discussed in recent conferences (PFAS Summit, Brussells, May 2025 and Fluoropolymers 25, Savannah, USA, June 2025).
Q4:The formation of the literatures should be paid attention, such as reference 84.
Answer 4: all references have been carefully checked and corrected.
Question 5: Figure 9 is not a key for the whole paper. So it should be deleted.
Answer: Following the referee’s request, Figures 7-10 have been deleted from the initial version.
Question 6 : Is closing the fluorine loop is a good solution for PFAS? The topic need more deep discussions. In my opinion, it is only a solution, not only ones.
Answer: related to « Closing the loop » (section 7), we have cited a recent article (Nature, 2025, 640, 100; ref.89) and our review (Chem. Soc. Rev, 2023, 52, 4208, ref. 88) that sums up that topic. Our point is that a possible solution for the PFAS-issue is essential with specific and sometimes High-Tech applications, closing the loop in manufacturing and uses and restrict/ban all the rest. It seems like that is pretty clear. Many other options are possible between doing nothing and full ban. We clearly choose our option. What is important is that the whole world does this, if not, what will Europe do with Chinese electrical cars, Japanese batteries, Taiwanese computers, and so on.
Reviewer 3 Report
Comments and Suggestions for Authors
Please see the attached letter.

Author Response
Question 1. This manuscript lacks of critical scientific analysis. It spent a long content to describe the PFAS, their catalog and the timeline of their development, and the application of PFAS. Those are just a descriptive overview. The manuscript did not provide the necessary summary of the literature of scientific findings and debates.
Answer 1: as explained in page 5, several examples of PFAS and non-PFAS have been supplied (Figure 2). Related to the criticism aspect (that we totally understand since this is requested for a review article), several parts and sentences have brought some criticims such as:
-Section 2, page 6 : Of the anthropogenic PFAS, the vast majority (about 65%) are non-functional and water-insoluble fluids and gasses, such as HCFC, HFC, HFE, HFO or PFC; many of these do not bioaccumulate in living organisms, but they are persistent by themselves or their degradation products.
-Section 2, page 6 : The origin is controversial, but degradation of man-made refrigerants, agrochemicals, drugs and fluoropolymer precursors are significant contributors to the estimated yearly increase of TFA and its salts of a few thousand ton per year.
-Section 2, page 7 : Man-made PFAS are expensive to manufacture resulting in a considerable environmental footprint
-Section 2, bottom of page 5 : Therefore, great care must be taken not to make generalizations or oversimplifications- made out of missing data on individual products- such as considering the big class as equal under the same PFAS-umbrella.
-Section 3, page 7 : it took until the beginning of the 20th century before organofluorine chemistry and its applications took off.
-Section 3, bottom line of page 7 : …is still the biggest volume man-made organofluorine compound.
-Section 3, page 8 : 3M Company acquired the technology and started development of commercial products, which came on the market starting 1952
-Section 4, page 10 : In view of these outstanding properties, finding PFAS-free alternatives for the multiple uses is and continues to be quite challenging.
-Section 4, pages 10-11 : but, because of the combination of relevant features of fluoropolymers and many practical limitations, no alternatives were proposed.
-Section 5, page 11 : The biggest growth markets are air conditioning & refrigeration, transportation, aluminum production, pharma and agrochemicals, construction, semiconductor and electronic applications.
-Section 5, page 11 : The potential is expected to increase in the future due to advancements in synthetic strategies to incorporate -CF3, -OCF3 or -SCF3 groups
Section 5, page 11 : Important health care applications for PFAS include medical devices
Section 5, page 13 : Interestingly, they are also food approved.
Section 5.4.2, page 14 : Although the market for oil-and water repellents is a specialty market, worldwide volumes were once ca. 20.000-ton products per year, now declining and being replaced by fluorine-free repellents…..as well as « as essential components in fire-fighting agents (AFFF and ATC). »
Section 6.3., page 16 : N-ethyl perfluorooctyl sulphonamido ethanol (EtFOSE) in weeks under aerobic conditions, but unfortunately leading to other persistent PFAS. On the other hand, some PFAS degrade slowly.
Section 6, bottom page 17 : They also reported new economical and sustainable treatment methodologies valuable to remove PFAS from the environment across the World. and then with a 100% closed fluorine loop in a short residence time (minutes) is available yet
Section 6, page 18 : However, no cost-effective solution for complete remediation (meaning capturing, full mineralization and recycling of the formed fluoride) with a 100% closed fluorine loop in a short residence time.
Section 6, page 18 : industry has taken featuring actions to reduce its environmental footprint.
Section 6, top page 19 : Air emissions have been drastically reduced by many producers, but further worldwide reductions are mandatory [88,92].
Section 6, page 19 : … although not always successful since most replacements are not drop-in solutions, are still in the R&D stage.
Section 6, page 19 : However, significant problems in extrapolating health effects from laboratory animals to humans in the context of PFAS exposure, remain
Section 6.5, page 19 : …although not always successful since most replacements are not drop-in solutions, are still in the R&D stage.
Section 6.6, page 19 : …has become the predominant sources of human exposure to PFAS.
Section 6.6a), page 20 : prove to have remarkable bioaccumulation in all living species, including the human body, and limited excretion by the kidneys
Section 6.6, page 19 : Interestingly, Gui et al. [99] reported an association between a PFAS exposure and risk of diabetes.
Section 6.6, page 19 : However, significant problems in extrapolating health effects from laboratory animals to humans in the context of PFAS exposure, remain [100].
Section 6.6.c), page 20 : Chronic toxicity and the underlying mechanisms for PFAS are still not very well understood.
Section 6.6.c), page 19 : although the liver toxicity in rodents can not be extrapolated to assess human health implications.
Section 6.6e), page 20 : Comparing the present and future concentrations of TFA [114], the dominant PFAS in the ecosphere, the threat for irreversible accumulation is rising [115], but risk for the environment and living species is anticipated to be under control for now [114,116]. However, a recent study raises.
Section 6.7, page 21 An important gap in analytical methods for individual compounds and sum parameters as well as sensitivity and cycle time (in-situ measurements) remains.
Section 7, page 21 : in our opinion, since PFAS contamination does not stop at borders and the fate of imports of PFAS-containing products into Europe is still unknown, the restrictions may not stop PFAS-contaminations in Europe and become a disruptive economic and strategic burden.
Section 7, last paragraph in page 21 : finding a new worldwide balance between the unique properties of PFAS and the environmental footprint on our global ecosphere and society, is an absolute must
Question 2: The language and the style of this manuscript read like an informal tone, towards a broad non-specialist audience rather than scientific peers. For example: the title of sub-title 6 “the future? Closing the fluorine loop!” , and line 133 “So, the question remains: “ A world without PFAS … Why (not)?”.
Answer : Efforts and improvements have been made to remove from an "informal tone" and the style has been improved. That review is dedicated to a wide audiance and we wanted to have all scientists to see the whole picture (including historical aspects) and therefore need to inform, educate and teach the correct and raised more information and evidences.
In addition, we have also improved several parts and sub-headings (such as the comments raised by the reviewer).
Question 3 : The overall logic of this manuscript focused on the advocacy and narrative coherence, rather than discussing the data or scientific findings, or the knowledge gap in this area. For example, this manuscript only gives a brief mention of the detection, monitoring, and remediation of PFAS, but without discussing or critically analyzing the analytical methods, the sensitivity or recent innovations. The authors should integrate more scientific data, comparative tables, and rigorous evaluation of the literatures and discuss the art-to-date research trends and scientific knowledge gap.
Answer: In pages 20-21, we have added a new section (6.7) related to analytical methods which shows that the scientists have a dilemma and a serious gap in knowledge. The conclusion has been reshuffled to expand on the gaps.
That review contains many data and Tables related to production, elimination, regulations, uses (e.g., the four groups, health care, food security, energy systems and materials, pages 11-12 and figures of fuel cell membrane for example) as well as treatment technologies of PFAS (pages 16-18) summarized by Figure 6.
Question 4: Inconsistent font for some content in this manuscript., e.g. line 40-51, line 308. While the manuscript covers a relevant and timely topic, its current form lacks the scientific depth and analytical rigor expected for publication as a review paper in Molecules. I believe that substantial revisions would be required to meet the standards of Molecules. It may be more appropriate as a perspective article, or science communication. Thank you again for the opportunity to review this submission.
Answer Related to the scientific depth and rigor, we have reshuffled and improved the revised manuscript inserting many yellow highlights and alos checked the font of the manuscript. We are satisfied that the reviewer consider that manuscript as a perspective article.
Reviewer 4 Report
Comments and Suggestions for Authors
I found the review insightful and well-structured, particularly in how it outlines the key challenges and critical issues related to PFAS contamination. The clarity in identifying the health and environmental impacts is commendable and offers a strong foundation for future research and policy development.
One aspect I would suggest expanding is the role of circularity as a strategic approach to mitigating PFAS contamination. While the review touches on regulation and remediation, the concept of circular economy — including sustainable product design, materials recovery, and the minimization of waste generation — appears to be underexplored. Given that circular strategies can significantly reduce the introduction and accumulation of PFAS in the environment, a more in-depth discussion in chapter 5 of this dimension would further strengthen the review and provide readers with a broader vision of integrated, long-term solutions.
Author Response
I found the review insightful and well-structured, particularly in how it outlines the key challenges and critical issues related to PFAS contamination. The clarity in identifying the health and environmental impacts is commendable and offers a strong foundation for future research and policy development.
One aspect I would suggest expanding is the role of circularity as a strategic approach to mitigating PFAS contamination. While the review touches on regulation and remediation, the concept of circular economy — including sustainable product design, materials recovery, and the minimization of waste generation — appears to be underexplored. Given that circular strategies can significantly reduce the introduction and accumulation of PFAS in the environment, a more in-depth discussion in chapter 5 of this dimension would further strengthen the review and provide readers with a broader vision of integrated, long-term solutions.
Answer: We thank the reviewer for delataury comments and we agree with her/him on such a matter. The circularity topic, also called «closing the fluorine loop» as another terminology, has been broadened in sections 6 and 7 (citing various key references including Nature, 640, 100 (2025); ref.89 and Chem. Soc. Rev., 2023, ref. 88). In addition, paragraphs in page 22 and Figure 9 (Life cycle and circularity of fluoropolymers) was already inserted in the initial version.
Round 2
Reviewer 2 Report
Comments and Suggestions for Authors
I feel it can be accepted in the current state.
Reviewer 3 Report
Comments and Suggestions for Authors
Thank you for addressing the previous comments and suggestions. After reviewing the revised manuscript, I find that the authors have made significant improvements in both content and clarity. The revised version addresses the major concerns raised in the initial review and presents a more coherent and well-structured discussion. I believe the manuscript is now suitable for publication in its current form.